# LLM-NAVI: NAVIGATING MOTION AGENTS IN CLUTTERED AND DYNAMIC ENVIRONMENTS VIA LLM REASONING

## ABSTRACT

We introduce **LLM-Navi**, a novel large language model-based (LLMs) framework for autonomous navigation in dynamic and cluttered environments. Unlike prior studies constraining LLMs to simplistic, static settings with limited movement options, LLM-Navi enables robust spatial reasoning in realistic, multi-agent scenarios, achieved by uniformly encoding the environments (e.g., real-world floorplans), dynamic agents, and their trajectories as *tokens*. In doing so, we unlock the zero-shot spatial reasoning capabilities inherent in LLMs without requiring retraining or fine-tuning. LLM-Navi supports multi-agent coordination, dynamic obstacle avoidance, and closed-loop replanning, demonstrating generalization across diverse agents, tasks, and environments through text-based interactions. Our experiments show that LLMs can autonomously generate collision-free trajectories, adapt to dynamic changes, and resolve multi-agent conflicts in real time. We extend this framework to humanoid motion generation, showcasing its potential for real-world applications in robotics and human-robot interaction. This work thus establishes a first foundation for integrating LLMs into embodied spatial reasoning tasks, offering a scalable and semantically grounded alternative to traditional methods.

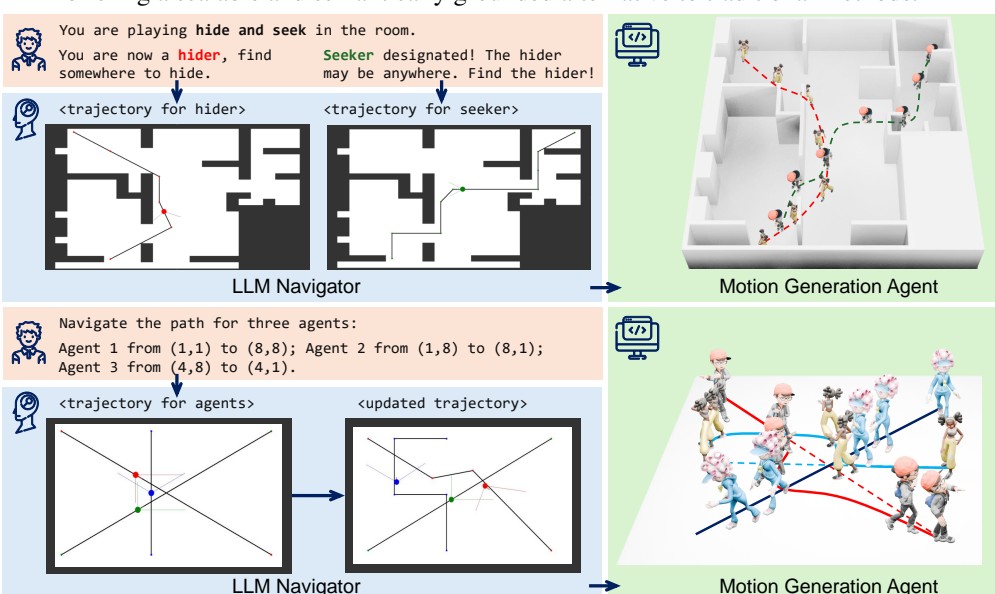

Figure 1: **Single agent** (top) navigating a real floor plan in a hide-and-seek scenario. **Multiple agents** (bottom) simultaneously move toward their destinations, the LLM autonomously resolves potential collisions. *LLM-Navi* is an autonomous path navigator with zero-shot spatial reasoning capabilities, effectively handling obstacle avoidance and collision resolution in dynamic environments.

## 1 INTRODUCTION

Large Language Models (LLMs) (Achiam et al., 2023; Team et al., 2024; Guo et al., 2025; Touvron et al., 2023) have made significant strides in reasoning and planning across various domains. These

models exhibit powerful generalization capabilities, enabling them to handle unseen scenarios with remarkable flexibility and comprehension of world knowledge (Sun et al., 2023; Wu et al., 2024c;a;b; Singh et al., 2023; Hendrycks et al., 2021; Sun et al., 2024). Despite their successes, the application of LLMs to spatial pathfinding and obstacle-free trajectory generation has received limited attention, particularly in real-world settings.

This paper proposes **LLM-Navi** to investigate the spatial reasoning ability inherent in pre-trained LLMs in zero-shot path navigation under dynamic and cluttered environments. As Figure 1 illustrates with a hide-and-seek and a three-agent example, our token–LLM pipeline converts plain-language goals into collision-free trajectories without any task-specific training. To make navigation tasks amenable to LLMs, akin to language tokens, we uniformly represent the environment (e.g., floor maps), agents, and paths as tokens that interact within a shared space. This token-based representation enables efficient spatial reasoning, providing robust and flexible solutions for pathfinding in complex environments. Specifically, we represent paths as sequences of sparse anchor points, allowing agents to move flexibly to any position within the space, rather than relying on predefined movement sets (e.g., 4 or 8 adjacent positions), as seen in previous works (Wu et al., 2024b; Yang et al., 2024; Aghzal et al., 2024; 2023; Martorell, 2025). This approach aligns more closely with human intuition for maneuvering and obstacle avoidance, while also reducing the processing burden on the LLM. Furthermore, we introduce two path refinement strategies, *additive* and *compositional*, which allow the system to self-correct trajectories without human-in-the-loop intervention and can be extended to dynamic settings where multiple agents coexist.

Classical planners excel once tuned to a single map, but retraining for every new layout or agent set is costly. A pretrained LLM, in contrast, can reason over any text token sequence; by expressing floorplans and agents in its native modality we unlock zero-shot spatial planning. Thus, this work represents a departure from the conventional approach in path navigation using reinforcement learning (RL) which, while performing well in controlled settings, requires extensive data collection and training and often struggles to generalize to novel scenarios—challenges where LLMs demonstrate promising zero-shot adaptability. Despite that, we believe LLM and RL are not at odds with each other; rather, their synergy provides complementary advantages. With the rise of LLMs, we believe the dataset and insight we develop in this paper will be very useful in extending cognitive and language reasoning into embodied spatial domains, offering a flexible and scalable alternative in settings where previous approaches face practical constraints.

To evaluate our approach, we construct a dataset from real floorplans and assess our training-free, LLM-powered system across various modern LLMs in both single-agent and multi-agent settings. Our results, e.g., Figure 1, show that this approach generalizes effectively to previously unseen environments and tasks. Additionally, we showcase the system's ability to solve dynamic problems in closed-loop environments, where agent communication and coordination are essential to ensure robust navigation and motion. Agents can autonomously adjust their plans and avoid collisions in real-time, highlighting the system's ability to handle complex, unpredictable scenarios. Finally, we showcase a proof-of-concept extension to humanoid motion generation, demonstrating how the system can generate realistic and contextually appropriate movements for agents in a wide range of dynamic environments. We believe this work will inspire real-world applications in areas such as autonomous robotics, virtual reality, and interactive human-robot interaction, where agents must navigate and collaborate in complex, dynamic spaces.

We summarize the contributions of **LLM-Navi** as follows:

- We introduce a unified token schema to represent environments, agents, and paths.
- We establish an evaluation protocol by constructing a dataset and defining standardized evaluation metrics.
- We demonstrate that our system achieves a 78% single-shot success rate, and our proposed refinement strategies boost performance to 87% and can sustain over 60% success with three interacting agents, highlighting strong potential for real-world applications.

## 2 RELATED WORK

**Spatial Understanding and Reasoning.** Effective navigation and planning in space require a fundamental understanding of the environment, a critical cognitive ability for both humans and

intelligent systems. With the advancement of LLMs, spatial reasoning has become an emerging research focus. Studies such as Wu & Guo; Mirzaee et al. (2021); Mirzaee & Kordjamshidi (2022); Momennejad et al. (2023) investigate LLMs' spatial reasoning capabilities through verbal reasoning tasks, such as question answering (QA). Other works like Mirchandani et al. (2023) explore LLMs' ability to recognize patterns, while Hong et al. (2023) focuses on enhancing LLMs' 3D reasoning capabilities. Additionally, Wu et al. (2024b) evaluates LLMs' performance in solving QA-based tiling puzzles. However, these studies largely focus on static or symbolic reasoning, and do not consider grounded, embodied spatial tasks that require interaction with dynamic environments.

We believe modern LLMs' powerful language reasoning can be extended into spatial reasoning with proper token representation. In this paper, we validate this idea in extensive quantitative and qualitative evaluations, demonstrating their effectiveness in understanding spatial environments and generating feasible solutions. Moreover, we showcase their downstream applications, such as humanoid motion, extending beyond theoretical or virtual scenarios to more practical settings.

**Using LLMs for Navigation.** Recent LLM studies on path navigation apply to restricted scenarios. For example, Wu et al. (2024b) proposes a protocol to evaluate LLMs' ability to navigate in simple environments with a single possible route; Aghzal et al. (2024; 2023) investigate LLMs' pathfinding capabilities in square grids with simplified, synthetic environments; Martorell (2025) explores using LLMs to navigate a single agent in a 5x5 grid without obstacles. All these studies focus on navigation within unreal environments with movements restricted to four directions (up, down, left, right), limiting their real-world applicability. Other works (Zhou et al., 2023; Yang et al., 2024; Liu et al., 2025; Yuan et al., 2024; Yu et al., 2023) employ language models for individual navigation tasks with camera inputs. However, these efforts are often limited to perception-conditioned action selection, without broader path planning, spatial negotiation, or multi-agent interaction.

In contrast, our approach enables global path planning over realistic environments with multiple agents, supporting continuous movement, interaction, and closed-loop adaptation. Unlike prior works constrained by low-dimensional grids or simple mazes, we demonstrate that LLMs can perform robust spatial reasoning in human-scale spaces, coordinating complex navigation tasks with no fine-tuning or retraining.

**Environment-Aware Agents.**

Path planning, especially for multiple interacting agents, is an active research area in robotics and AI, commonly studied as Multi-Agent Path Finding (MAPF) (Stern et al., 2019; Li et al., 2021; Shaoul et al., 2024). Classical planning methods, such as the A* algorithm (Hassan et al., 2021; Zhao et al., 2023), effectively solve single-agent navigation tasks in static environments but often face challenges in dynamic or multi-agent scenarios due to agent interactions and collision avoidance requirements. Recently, learning-based approaches—including diffusion models (Yi et al., 2024; Rempe et al., 2023; Liang et al., 2024) and reinforcement learning (Chao et al., 2021; Peng et al., 2022; Hassan et al., 2023)—have also been explored for complex navigation tasks. However, these typically require task-specific training and extensive data collection, posing limitations on their zero-shot generalizability and deployment flexibility. In contrast, our work explores a training-free approach, leveraging the zero-shot spatial reasoning and interactive planning capabilities of LLMs. Our method is complementary to existing MAPF and learning-based frameworks, focusing explicitly on LLMs' inherent capabilities for dynamic multi-agent navigation tasks through intuitive, human-like interaction and iterative refinement.

## 3 Dynamic Path Planning with LLM-Navi

To make LLMs comprehend this task, we uniformly encode agents with their anchored trajectories and the environment as discrete text tokens interacting with each other as follows:

### 3.1 Agents with Anchored Trajectories

Echoing how humans navigate through their environment, we are not restricted to only a few discrete directions. Instead, human paths resemble a sequence of anchor points connected by almost straight lines to avoid obstacles while minimizing the total travel distance. For instance, when someone moves from one room to another, they may first walk directly toward the door, then proceed straight

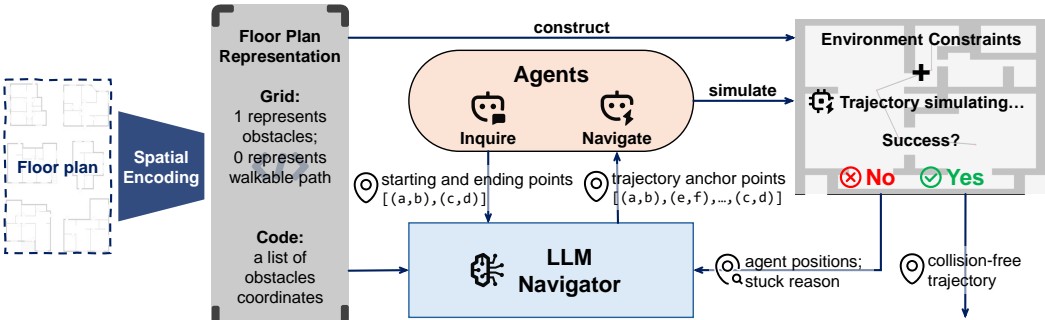

Figure 2: **LLM-Navi pipeline.** The process begins with the input of a floor plan, followed by spatial encoding, agent navigation, and simulation. LLM-Navi generates the agent's path, which is then validated and refined. If agents become stuck, they communicate with the LLM for guidance to resolve navigation issues.

to the door of the next room, and finally enter the target room. This anchor-based approach is intuitive for humans and can be easily adapted for LLM planning. In this framework, a path or trajectory is not necessarily dense; it only requires a sequence of key points, akin to decomposing a complex task into simpler sub-goals, which is well-suited for modern LLMs (Sun et al., 2023; Wu et al., 2024c; Wang et al., 2023; Prasad et al., 2023; Singh et al., 2023; Wu et al., 2024a; Wang et al., 2024).

Formally, let $\mathcal{X}$ denote the movable space of the agent, where each point $x \in \mathcal{X}$ represents a possible state of the agent. A trajectory $\mathcal{T}$ is defined as a sequence of points:

$$\mathcal{T} = \{x_1, x_2, \ldots, x_k\},$$

where each point $x_i \in \mathcal{X}$ represents a position in the environment, and $k$ is the number of anchor points in the path. The trajectory is generated by connecting these points with straight lines, and the travel distance can be formulated as:

$$D(\mathcal{T}) = \sum_{i=1}^{k-1} \|x_i - x_{i+1}\|_2,$$

where $\|\cdot\|_2$ denotes the Euclidean distance between two consecutive points. The task of path planning is to determine an optimal sequence of anchor points that connect the initial and goal states, while satisfying any environmental constraints.

A critical aspect of path planning is ensuring that the trajectory avoids obstacles, static or dynamic. Specifically, a sub-path between two consecutive anchor points $x_i$ and $x_{i+1}$ is valid if and only if there are no obstacles along the line segment connecting them. Formally, for each consecutive pair of points $(x_i, x_{i+1})$, the sub-path is valid if:

$$\forall t \in [0, 1], \text{ such that } \gamma(t) = (1 - t)x_i + tx_{i+1},$$
$$\text{there exists no obstacle such that } \gamma(t) \in \mathcal{O},$$

where $\gamma(t)$ represents the linear interpolation between $x_i$ and $x_{i+1}$, and $\mathcal{O}$ denotes the set of obstacles in the environment. If this condition is satisfied for all consecutive pairs $(x_i, x_{i+1})$, the trajectory $\mathcal{T}$ is considered valid.

Thus, the path planning problem can be viewed as finding an optimal sequence of anchor points $\{x_1, x_2, \ldots, x_k\}$ that minimize the total travel distance $D(\mathcal{T})$, while ensuring the trajectory avoids obstacles and adheres to other environmental constraints.

In multi-agent scenarios, the complexity of path planning increases, as the trajectories of different agents may intersect. While some intersections may be unavoidable or difficult to resolve, it is important to note that different agents occupy the same space at different times. Therefore, the actual trajectory of each agent can only be fully determined during testing, when the agents' interactions and timing can be accounted for in the dynamic environment.

### 3.2 SPATIAL ENVIRONMENT REPRESENTATION

Among various alternatives, the most commonly used and intuitive representation of space is in the form of grids (Yang et al., 2024; Wu et al., 2024b; Aghzal et al., 2024). In this representation, the

environment is discretized into a grid structure where each cell corresponds to a specific location, allowing for clear and precise definitions of free spaces, obstacles, and agent positions.

Alternatively, space can be represented using code-based descriptions (Aghzal et al., 2024), which can be more interpretable for LLMs. This approach is both compact and flexible, enabling precise definitions of the environment through code. For instance, variables can be defined to specify the start and goal locations, while logic can be applied to place obstacles on the grid, shaping the environment accordingly. Intuitively, code provides a clear and concise way to define the task setting, making it a powerful alternative to traditional grid-based representations.

By using text-based representations, we bridge the gap between spatial reasoning and natural language processing, enabling LLMs to leverage their reasoning capabilities in a domain where they have proven effectiveness. This text-based approach sets us apart from images as input, which can introduce unnecessary or redundant information such as textures and colors, while enabling the zero-shot reasoning capabilities of LLMs on language tokens similarly on uniform discrete tokens representing the environment, dynamic agents, and their trajectories.

To formalize, we define a grid-based environment representation:

$$\mathcal{G} = \{g_{i,j} \mid g_{i,j} \in \{0,1\}\}$$

where $g_{i,j} = 1$ indicates an obstacle, $g_{i,j} = 0$ denotes free space, and $g_{i,j}$ represents the cell at row $i$ and column $j$ in a 2D grid.

We also define a code representation as a list of obstacle coordinates:

$$\mathcal{C} = \texttt{obstacles.append}((i_1, j_1), \ldots, (i_n, j_n))$$

where each $(i_k, j_k)$ denotes the location of an obstacle.

### 3.3 SYSTEM ARCHITECTURE

#### 3.3.1 OVERVIEW

The architecture of the LLM-centered system is shown in Figure 2. Let the floor map be firstly encoded into a grid-based $\mathcal{G}$ or code-based $\mathcal{C}$ format. This encoded floorplan constructs an environment $\mathcal{E}$, and is passed to the LLM $\mathcal{L}$.

Given $N$ agents, each agent $i$ has a starting point $s_i$ and a target point $t_i$, LLM-Navi generates the trajectories $\mathcal{T}_i$ for all agents based on the starting points $S = \{s_i\}_{i=1}^N$ and target points $T = \{t_i\}_{i=1}^N$:

$$\mathcal{T} = \mathcal{L}(\mathcal{E}, S, T) = \{x_1^{(i)}, x_2^{(i)}, \ldots, x_{k_i}^{(i)}\}_{i=1}^N$$

The agents are simulated in environment $\mathcal{E}$. If a collision occurs at time $t$, the agent queries the LLM using its current position $\mathbf{p}_i(t)$ and the set of detected collisions $\mathcal{C}_t$, requesting a refined path $\mathcal{T}_i'$.

The final output consists of the collision-free trajectories $\mathcal{T}_i'$ for all agents, ensuring that each agent reaches its target point without collisions.

#### 3.3.2 PATH REFINING STRATEGIES

To handle collisions with static obstacles (floorplan) or dynamic obstacles (motion agents), we capitalize LLMs' multi-turn capability, allowing for iterative refinement outputs until achieving the desired result. We propose two transformative update strategies to refine the path: (1) *additive* and (2) *compositional*, drawing inspiration from image alignment warping source to target (Baker & Matthews, 2004). The additive approach recalculates the entire motion plan holistically, integrating all prior adjustments into a unified transformation—similar to continuously recalculating an optimal golf swing based on previous strokes. While straightforward, this method can be inefficient, as each update effectively resets to the origin. In contrast, the compositional approach refines the trajectory incrementally, making step-by-step corrections based on the current state. Though generally more efficient, it may suffer if an update places the trajectory in an unfavorable position (e.g., low terrain where the ball becomes stuck), potentially hindering future corrections.

In our setting, let $s$ denote the starting point and $t$ the destination. Given $n$ update opportunities, if an agent becomes stuck during the $i$-th trial for a reason $r$ (which includes its current stuck position),

the *additive* strategy updates the trajectory as

$$\mathcal{T}_{i+1} = \mathcal{L}(s, t, r),$$

implying that each update is computed globally, restarting from the original starting point $s$.

In contrast, the *compositional* strategy refines the trajectory based on the current state. Let $p_i$ denote the current stopping position at iteration $i$ when the path is unsuccessful. Then, the updated trajectory is computed as

$$\mathcal{T}_{i+1} = \mathcal{L}(p_i, t, r),$$

indicating that the current stuck position serves as the new starting point for planning the remainder of the path. Each corrective adjustment is applied "on-the-fly" to the current trajectory, enabling dynamic, incremental updates.

When multiple agents become stuck simultaneously, the LLM can coordinate across them using either strategy to refine their paths.

## 4 EXPERIMENTS

### 4.1 EXPERIMENT SETUP

#### 4.1.1 DATASET

To ensure the applicability of our problem to real-world scenarios while maintaining simplicity, we build on the R2V dataset (Liu et al., 2017), which contains 815 realistic floorplans from actual buildings. For each floor plan, we first convert it into textual formats and randomly sample three pairs of starting and target points. We then use the A* algorithm to generate obstacle-free optimal paths as ground truth labels, creating a dataset suitable for evaluating the spatial navigation ability of LLMs. In addition to this dataset, we manually created more complex scenarios to explore the upper bound of LLMs' navigation and planning capabilities, inspiring broader exploration in this domain.

#### 4.1.2 EVALUATION METRICS

We evaluate our method in a simulated environment[1]. We use several standard metrics: Success Rate (SR), Success weighted by Path Length (SPL), Completion Rate (CR), and Weighted Success Rate (WSR).

**Success Rate (SR)**: The share of evaluation episodes in which the agent actually reaches the goal.

**Success weighted by Path Length (SPL)**: Starts with the success rate, but then rewards agents that take near-optimal routes and penalises those that wander: a successful run counts most when its path is as short as the best possible one.

**Completion Rate (CR)**: Looks only at how much of the planned route the agent covers. Even if it never gets to the target, it earns partial credit for the fraction of the path it did traverse.

**Weighted Success Rate (WSR)**: Counts successes, but gives more weight to episodes whose optimal paths are longer (and therefore usually harder). Finishing a difficult, long-distance task contributes more than finishing an easy, short one.

These metrics, described in detail in the Appendix A, are commonly used in navigation and path planning tasks to assess both the efficiency and effectiveness of the trajectory generation. They offer a comprehensive assessment of LLMs' navigation performance by considering success rates, trajectory efficiency, task completion, and complexity. Using these metrics in conjunction with our constructed dataset, we propose a protocol to benchmark the spatial navigation capabilities of LLMs.

### 4.2 QUANTITATIVE RESULTS

We evaluate a range of LLMs, including GPT-4o (Hurst et al., 2024), Gemini (Team et al., 2023), DeepSeek (Guo et al., 2025), Llama (Dubey et al., 2024), OpenAI o3-mini, and Claude-Sonnet. Our

---

[1]Unlike Wu et al. (2024b), we do not evaluate token overlap, as two anchor lists that differ syntactically may still yield identical trajectories.

selection comprises both state-of-the-art reasoning models and prior general-purpose models. We conduct extensive experiments, including benchmarking and ablation studies. All of our experiments are in a *zero-shot* setting.

### 4.2.1 SINGLE-AGENT

First, we assess how LLM-Navi performs in a single attempt. As a baseline, we use scores obtained by moving directly from the starting point to the destination without any navigation. For input, we examine both grid and code representations, as described above. Additionally, we explore direct image input without spatial encoding, where the start and end points are marked on the image.

The results, presented in Table 1, indicate that modern models with advanced reasoning capabilities demonstrate significantly stronger performance in spatial navigation tasks. Furthermore, textual input outperforms image-based input, highlighting the effectiveness of textual representations, which align more naturally with LLMs' ability to process structured, discrete text-based information. When comparing the two textual formats, their effectiveness varies across models. For most models, code-based input yields better performance, as it explicitly encodes coordinates. However, for o3-mini, which exhibits strong reasoning capabilities, the grid-based format proves more effective. We attribute this to its ability to intuitively recognize spatial patterns, akin to human interpretation.

We also investigate how the LLM-based system benefits from multi-turn interactions. We evaluate and compare the *additive* and *compositional* approaches, with results shown in Figure 3. As the number of turns increases, both methods contribute to overall performance improvement. The *additive* approach generally achieves a higher success rate (both SR and WSR), as it recalculates the complete path from the origin at each step. In contrast, the *compositional* approach, while more susceptible to suboptimal adjustments—analogous to a poor stroke in golf affecting subsequent corrections—exhibits higher SPL and CR, as it refines the current trajectory without resetting, preserving progress and ensuring continuous improvement.

| Models | Type | Input | SR ↑ | SPL ↑ | CR ↑ | WSR ↑ |
|---|---|---|---|---|---|---|
| *Baseline* | - | - | 0.370 | 0.370 | 0.370 | 0.207 |
| Claude-3.5-Sonnet | general | code | 0.453 | 0.437 | 0.556 | 0.307 |
| | | grid | 0.330 | 0.302 | 0.426 | 0.183 |
| Llama-3.3-70B | general | code | 0.397 | 0.316 | 0.565 | 0.240 |
| | | grid | 0.350 | 0.296 | 0.520 | 0.197 |
| Gemini-2.0-flash | general | code | 0.380 | 0.297 | 0.541 | 0.205 |
| | | grid | 0.273 | 0.223 | 0.328 | 0.139 |
| GPT-4o | multimodal | code | 0.420 | 0.381 | 0.560 | 0.291 |
| | | grid | 0.387 | 0.376 | 0.477 | 0.227 |
| | | image | 0.370 | 0.349 | 0.477 | 0.222 |
| DeepSeek-R1 | reasoning | code | 0.673 | 0.645 | 0.766 | 0.520 |
| | | grid | 0.650 | 0.623 | 0.729 | 0.503 |
| o3-mini | reasoning | code | 0.507 | 0.488 | 0.590 | 0.396 |
| | | grid | 0.781 | 0.710 | 0.828 | 0.665 |

Table 1: Zero-shot path navigation performance of various LLMs for single-agent scenarios in a single trial. Underlined numbers are significantly higher than *every* other model in the same column under a paired two-sided statistical test ($p<0.05$).

### 4.2.2 MULTI-AGENTS

We extend our experiments to scenarios with two and three agents, with the results shown in Figure 4. In these experiments, the maximum number of retries is set to three. This setup introduces additional complexity, as each agent must navigate to its destination while avoiding obstacles and dynamically resolve collisions with other agents, making the system more interactive and adaptive.

Our results indicate that as the number of agents increases, the performance scores decrease, but they remain within a reasonable and practical range. This demonstrates the feasibility of multi-agent coordination and LLMs' ability to manage complex, dynamic environments. Additionally, different update strategies show minimal impact on overall performance. The *compositional* approach tends to yield slightly lower scores, which aligns with real-world observations—for instance, when two people walk toward each other and both instinctively step in the same direction to avoid a collision, they may inadvertently create an awkward situation. This suggests that the *additive* approach may be more effective in globally resolving such coordination conflicts.

Overall, modern LLMs, such as o3-mini, demonstrate substantial capabilities in spatial navigation tasks, achieving approximately 80% SR in a single trial and up to about 90% SR with multiple trials for single-agent scenarios. The system can also be seamlessly extended to multi-agent scenarios without much performance degradation. Moreover, all experiments are conducted in a zero-shot setting, further validating LLMs' reasoning abilities and world knowledge. Taken together, these results indicate that modern LLMs are becoming increasingly applicable to real-world agent scenarios.

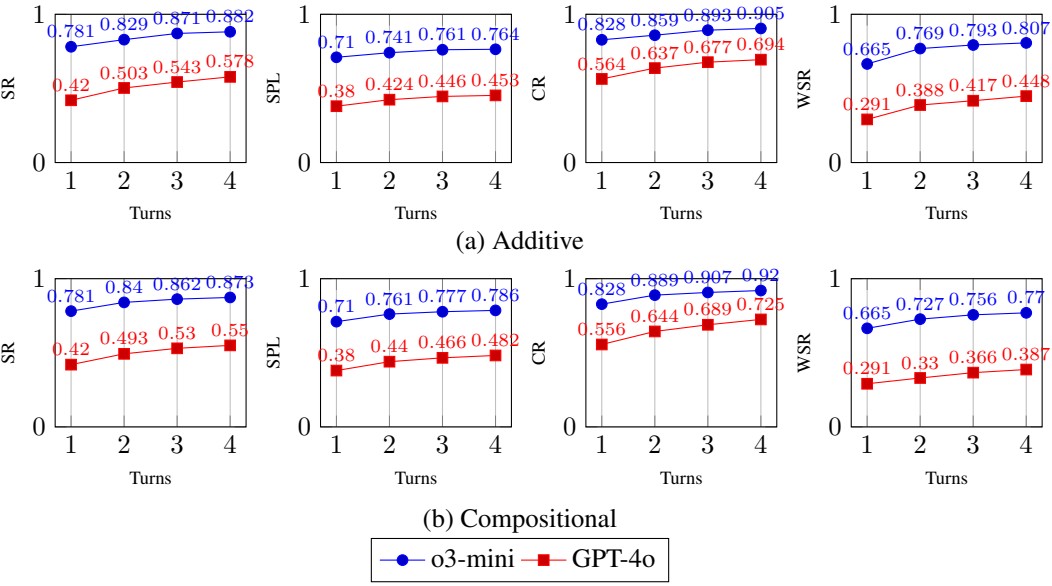

(a) Additive

(b) Compositional

Figure 3: Additive and compositional strategies for multi-turn navigation refining.

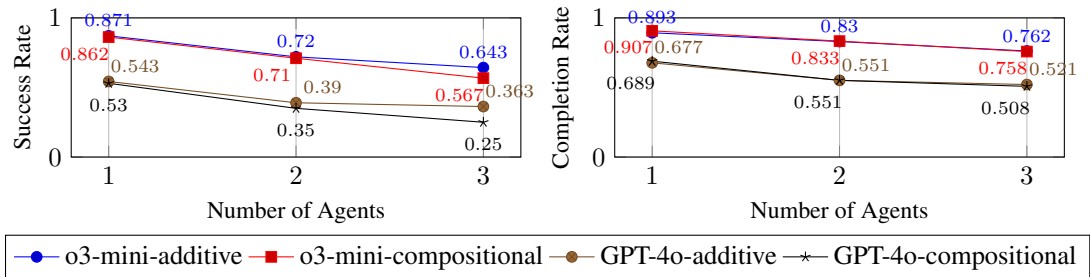

Figure 4: Success Rate (SR) and Completing Rate (CR) for different numbers of agents.

### 4.3 QUALITATIVE RESULTS

In this section, we visualize our results and introduce a practical application involving generating environment-aware humanoid motion (Figure 5a). Previous works, such as OmniControl (Xie et al., 2023) and TLControl (Wan et al., 2024), support trajectory control; however, they rely on *manually* defined input trajectories. Frameworks like Motion-Agent (Wu et al., 2024a) leverages LLMs to automatically decompose complex user requests and generate motion through a motion generation agent, enabling natural user-agent interaction. By integrating these approaches, we demonstrate that our LLM-based navigation system can be seamlessly applied to humanoid motion as a downstream task. Furthermore, our approach can be readily extended to scenarios involving multiple agents coexisting within an environment. This application method is illustrated in Figure 5a. Once the system generates an obstacle-free trajectory, it can be used to guide motion generation models — which are inherently non-environment-aware — to follow the trajectory and thus avoid collisions.

We provide both the top-down view and 3D humanoid motions in Figure 6, demonstrating that our LLM-based system can navigate effectively for both single and multiple agents. While the initially generated path may sometimes be infeasible, the system can autonomously adjust itself multiple times to resolve such issues. Notably, in the example on the right, where five agents encounter different challenges, the system successfully coordinates their paths through multiple adjustments, enabling them to avoid obstacles and one another, ultimately reaching their respective destinations.

We extend the input floor map to 3D (or more precisely, 2.5D), where each point is assigned a height value, forming a height map. Consequently, the output path is represented using anchor points in three dimensions. As shown in Figure 5b, this extension enables our system to generate agent paths that are not restricted to a flat plane. Additionally, our system has the potential to be integrated with methods like SCENIC (Zhang et al., 2024), which can interact with the scene but rely on manually defined input trajectories to avoid obstacles.

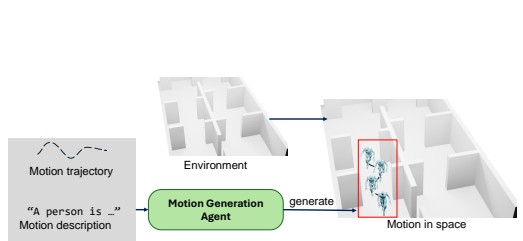

Agent walks up the stairs from (1,4,0) to (5,4,1).

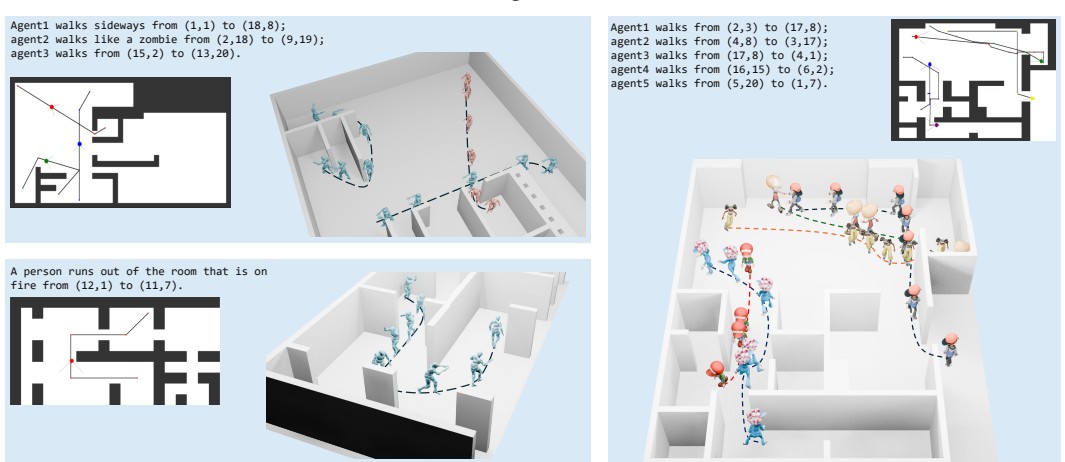

(a) When paired with motion generation agents LLM-Rav can generate environment-aware, realistic motions.

(b) LLM-Rav generates a 3D path and guides the blue agent to move within the 3D space while avoiding the red agent intruding during execution by adjusting the path.

Figure 5

Figure 6: Top-down views of the generated final trajectory and visualized humanoid motion results. Agents successfully navigate to their intended destinations while avoiding obstacles and other agents.

## 5 DISCUSSION

**Limitations and Future Work.** Our first work on LLM-navigation focuses on dynamic systems, including multi-agent scenarios, and has been validated using a simulated environment. Although these simulations offer valuable insights into system effectiveness, the lack of real-world testing—particularly with physical robots and in household scenarios—necessitates further validation in real-world settings. Additionally, our current approach relies on a globally encoded floorplan that assumes full observability during path planning. Future work will explore replacing the global floorplan embedding with a local embedding strategy that focuses on the immediate surroundings observable by each agent at the current timestamp. Nevertheless, our framework is inherently generalizable, capable of seamlessly incorporating additional functionalities such as collision handling and agent-agent interactions, which can be managed as local operations.

**Concluding Remarks.** In this work, we examined the reasoning capabilities of large language models (LLMs) in spatial navigation and collision-free trajectory generation for motion agents in dynamic environments. We represent floor maps as discrete text and structure navigation paths using a sparse anchored representation. As LLM-Navi is one of the first studies on LLMs' spatial reasoning, we constructed a comprehensive dataset and proposed evaluation protocols to assess their performance. Furthermore, we extended our investigation to scenarios involving multiple coexisting agents. Our results demonstrate that LLMs can effectively coordinate across agents and autonomously resolve collisions in closed-loop, dynamic settings. We showcased the real-world applicability of our approach by applying it to the task of humanoid motion.

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

APPENDIX

## A  EVALUATION METRICS

In this section, we detail the four standard metrics (see 4.1.2) used to evaluate the performance of LLMs: Success Rate (SR), Success weighted by Path Length (SPL), Completion Rate (CR), and Weighted Success Rate (WSR).

**Success Rate (SR):** The success rate is defined as the percentage of test cases where the agent successfully reaches the goal:

$$SR = \frac{1}{N} \sum_{i=1}^{N} \mathbb{I}(\text{success}_i),$$

where $\mathbb{I}(\text{success}_i)$ is an indicator function that returns 1 if the agent successfully reaches the target and 0 otherwise, and $N$ is the total number of test cases.

**Success weighted by Path Length (SPL):** SPL accounts for both the success rate and the efficiency of the path:

$$SPL = \frac{1}{N} \sum_{i=1}^{N} \frac{\mathbb{I}(\text{success}_i) \cdot d_i}{\max(d_i, d_{\text{opt},i})},$$

where $d_i$ is the length of the trajectory taken by the agent, and $d_{\text{opt},i}$ is the optimal path length for the $i$-th test case. The SPL metric rewards shorter, more efficient paths while penalizing longer, inefficient paths.

**Completion Rate (CR):** The completion rate measures the fraction of the total path length that the agent is able to complete. It is defined as:

$$CR = \frac{1}{N} \sum_{i=1}^{N} \frac{d_i}{d_{\text{total},i}},$$

where $d_i$ is the length of the trajectory taken by the agent, and $d_{\text{total},i}$ is the total length of the path the agent was supposed to cover in the $i$-th test case. The CR metric emphasizes how much of the planned path is successfully completed, regardless of success or failure.

**Weighted Success Rate (WSR):** WSR is a metric that assigns higher weights to longer paths, reflecting their cost or complexity, defined as:

$$WSR = \frac{1}{\sum_{i=1}^{N} d_{\text{opt},i}} \sum_{i=1}^{N} \mathbb{I}(\text{success}_i) \cdot d_{\text{opt},i},$$

where $d_{\text{opt},i}$ is the length of the optimal path, which can also reflect the difficulty of the test case. The denominator normalizes the WSR across all test cases, ensuring that the sum of WSR equals 1 by considering the total optimal path length.

## B  MORE QUALITATIVE RESULTS

We provide additional qualitative results in the supplementary material, including attached videos and an HTML file for better visualization.

Here, we present some final generated trajectories in top-down views.

Figure 7 illustrates the capability of the LLM to effectively manage and resolve complex and challenging scenarios.

Figure 8 demonstrates how the additive strategy utilizes a restart mechanism to successfully avoid obstacles.

Figure 9 demonstrates how the compositional strategy effectively dynamically avoid obstacles "on the fly".

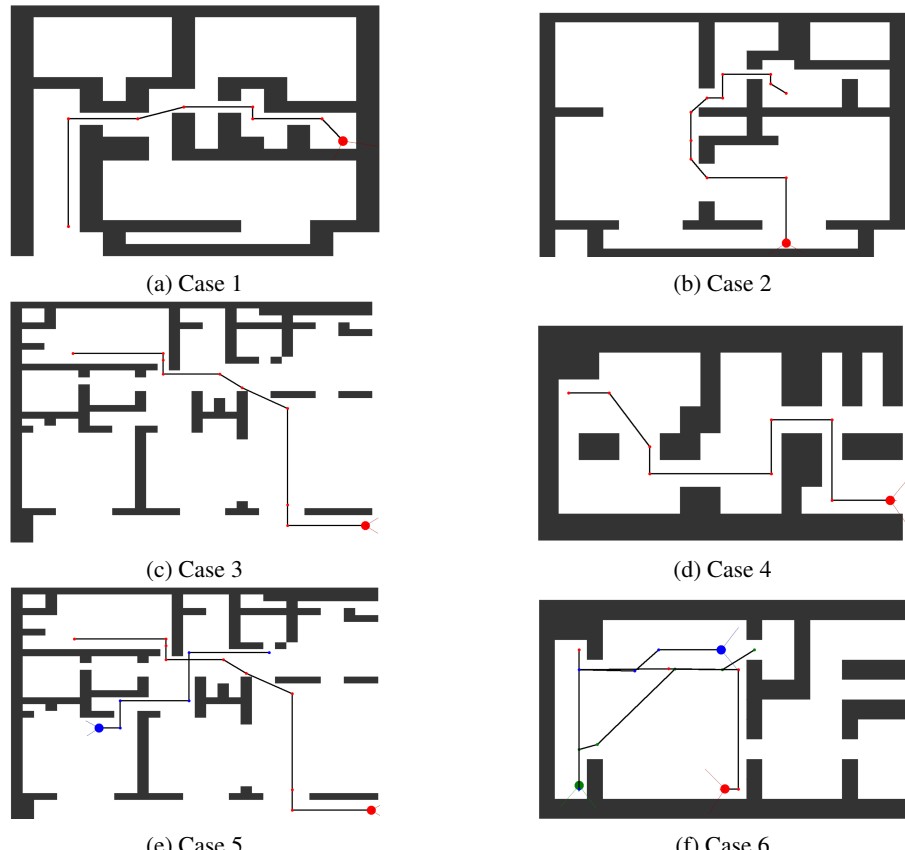

(a) Case 1  (b) Case 2

(c) Case 3  (d) Case 4

(e) Case 5  (f) Case 6

Figure 7: More qualitative results in top-down view.

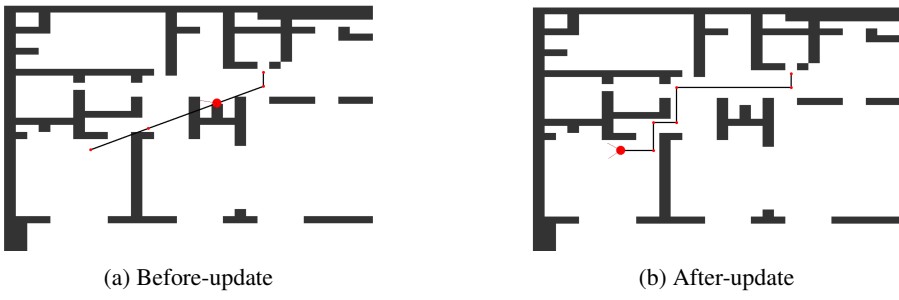

(a) Before-update  (b) After-update

Figure 8: Additive Strategy

## C  DATASET PROCESSING

### C.1  SPATIAL ENCODING

Given an input floor plan image $I \in \mathbb{R}^{H \times W \times 3}$, we first convert it to a grayscale image:

$$I_g = \text{Grayscale}(I), \quad I_g \in \mathbb{R}^{H \times W}.$$

Next, we remove all entirely black rows and columns to extract the significant region $I' \subseteq I_g$:

$$I' = I_g[\text{rows}(I_g) \neq 0, \text{ cols}(I_g) \neq 0].$$

We then pad $I'$ appropriately to standardize its dimensions and apply resizing to reduce the spatial resolution, yielding $I_r$. To enhance structural coherence, Gaussian smoothing is performed:

$$I_s = \text{GaussianBlur}(I_r).$$

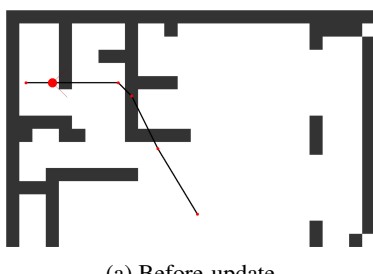 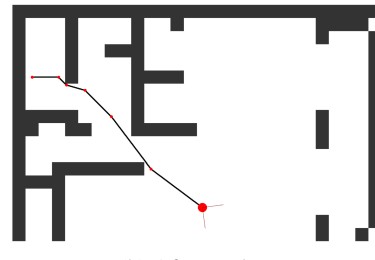

(a) Before-update          (b) After-update

Figure 9: Compositional strategy

Finally, we binarize the image $I_s$ to produce a binary representation $I_b$:

$$I_b(x, y) = \begin{cases} 1, & I_s(x, y) \neq 0 \\ 0, & \text{otherwise} \end{cases}.$$

The resulting binary matrix $I_b$ represents navigable spaces and obstacles explicitly, serving as input for subsequent spatial reasoning tasks.

### C.2 SAMPLING

To increase task complexity beyond uniform selection, we employ a strategy for sampling the start and target positions. For a given start cell $s$ and any candidate cell $c$, the Manhattan distance is defined as

$$d(c) = |i_s - i_c| + |j_s - j_c|.$$

Let $d_{\min}$ and $d_{\max}$ be the minimum and maximum distances from $s$ to all candidate cells, respectively, and normalize the distance as

$$\hat{d}(c) = \frac{d(c) - d_{\min}}{d_{\max} - d_{\min}},$$

with the convention $\hat{d}(c) = 0$ when $d_{\max} = d_{\min}$.

Given a distance weight $\alpha \in [0, 1]$, we compute the weight for each candidate as

$$w(c) = \alpha \hat{d}(c) + (1 - \alpha).$$

Then, the probability of selecting cell $c$ is

$$P(c) = \frac{w(c)}{\sum_{c' \in C} w(c')},$$

where $C$ is the set of candidate cells. This approach biases the selection towards cells farther from $s$ as $\alpha$ increases, while still preserving an element of randomness. In our implementation, we set $\alpha = 0.5$.

## D MORE DISCUSSION

**Comparison with SOTA.** It is worth noting that in the NeurIPS 2024 paper Wu et al. (2024b), the tasks are significantly simpler, involving small grid maps with only a single possible route and support for only four movement directions. This setting can be considered a strict subset of our case. Despite the simplicity, their reported navigation SR and CR using GPT-4 are only around 15% and 40%, respectively. We attribute this to the lack of a well-structured and standardized task formulation. Their outputs often contain additional words beyond the intended answer, requiring substring matching for evaluation. In contrast, our task employs a formalized output format, akin to those used in modern LLM-based planning agents, ensuring a more standardized and structured approach to path generation, and hence can be more easily applied to real-world scenarios.

**RL-agents vs LLM-agents.** While modern LLMs are not at odds with RL as they are trained with RLHF Achiam et al. (2023); Ouyang et al. (2022), in stark contrast to conventional deep RL in robot path planning which stresses on optimizing expected discounted return (for single-agent RL) and the equilibrium of joint policies (in multi-agent RL), typically with considerable amount of training data, this paper demonstrates the zero-shot, training-free ability across modern LLMs in path planning and dynamic navigation in single-agent and multi-agent scenarios, evaluated on completion rate, success rate and path length which, albeit their simplicity and lack of sophistication compared to the specific and formal RL optimization objectives, are arguably equally relevant to any autonomous systems in evaluating their performance.

Given sufficient training data, the ballmark success rate (SR) for deep RL in simulated environments may exceed 0.9 with proximal policy optimization and soft actor-critic algorithms, depending on the complexity of the environment, which may be cluttered with other dynamic obstacles. Our zero-shot SR and related performances, as shown in Figures 3 and 4 in the main paper, can reach this performance with typical performances above 75% in multi-turn navigation.

With no training data, we have the same performance because we are zero-shot, while the ballmark SR for deep RL will drop close to zero when it relies on exploration to learn optimal policies, which can be extremely time-consuming to even discover basic navigation strategies.

Notwithstanding, as modern LLMs are trained with reinforcement learning from human feedback (RLHF), it will be an interesting and worthwhile future work to incorporate deep RL, now possible with much less training data, into our work, to gain deeper insight on LLM and deep RL integration for dynamic path navigation while further improving performance.

