# OpenReview forum: "LLM-Navi: Navigating Motion Agents in Cluttered and Dynamic Environments via LLM reasoning"
_ICLR.cc/2026/Conference — ICLR 2026 Conference Withdrawn Submission_

### Official Review · Reviewer_Cq4W · 2025-11-01

**Soundness:** 2
**Presentation:** 3
**Contribution:** 1
**Rating:** 2
**Confidence:** 3

**Summary:**

The paper introduces LLM-Navi, an algorithm that prompts LLM to solve indoor (multi-agent) navigation tasks. LLM-Navi encodes the map into text with code-based or grid-based representations, then queries an LLM to provide a solution and iteratively refine the path in case of collision. The paper also demonstrates that LLM-Navi-generated paths can be combined with motion generation methods for downstream tasks such as humanoid path planning.

**Strengths:**

- The writing is overall clear, barring some notation issues.
- The proposed method is novel to my knowledge.
- The empirical evaluations show that LLM-Navi improves significantly over naive baselines.
- The video demonstrations in the supplementary materials are helpful.

**Weaknesses:**

- This paper reads like a compilation of design choices to make LLM-assisted path finding work without much insight or reasoning behind them. The design choices include map representations (grid/code), refining strategies (additive/compositional), and the LLM models. The authors tested all the possible combinations and made conclusions directly from the results. I am not sure if a reader can learn beyond the very scope of this paper.
    - Branching off this argument, specifically the additive and compositional refining strategies. It seems like the only difference is whether to regenerate the trajectory from the collision point or from the start. If this is the case, I don't think this deserves a whole subsection to introduce. An interesting point mentioned in the paper is that the additive strategy allows on-the-fly refining, but this point was not developed further.

- The lack of good baselines is concerning. The only baseline is a naive solution that completely ignores the map. If there are no good prior works in this domain to compare to, at least create a decent heuristic solution.
    - Building on this, what is the advantage of using LLM-Navi instead of A*?

- There are no discussions on the LLM prompts, which I presume is an important part of the algorithm. No LLM templates or examples are provided either, which hinders the reproducibility of the paper.

- In the paragraph starting from L421, the author claims that LLM-Navi can readjust and solve the collisions. I would appreciate a deeper dive into how LLM-Navi solves these challenges. An interaction log would be helpful.

- It's not clear to me how LLM-Navi helps downstream motion generation since LLM-Navi treats humanoids also as point agents. It seems to me that these are two independent components.

**Questions:**

- L200, is it supposed to be "there exists no $t$ such that ..." since $\mathcal{O}$ is the set of obstacles?
    - L195 mentioned dynamic obstacles, which are not captured in this formulation and are not developed further in the paper.

- L272, $s$ and $t$ here means starting and target position, but $t$ is previously used to denote time (L252, L200). The starting and target position was $s_i$ and $t_i$ for agent $i$ in L248, but here $i$ becomes the $i$-th iteration. Consider using a different variable for iteration number and keep $i$ for agent.

- Is paragraph starting from L373 supposed to be in a separate conclusion section? Currently it's in a section discussing multi-agent scenarios.

- Figure 5, "LLM-Rav" -> "LLM-Navi"

---

### Official Review · Reviewer_XCoe · 2025-11-01

**Soundness:** 3
**Presentation:** 3
**Contribution:** 3
**Rating:** 6
**Confidence:** 2

**Summary:**

This paper proposed a novel framework that leverages pre-trained LLMs for autonomous navigation in dynamic and cluttered environments, enabling robust zero-shot spatial reasoning by uniformly encoding environments, dynamic agents, and their trajectories as discrete text tokens. Evaluated on a dataset built from 815 real floorplans, LLM-Navi shows strong performance.

**Strengths:**

1. This paper demonstrates notable originality across multiple fronts, redefining how LLMs interact with spatial navigation tasks.
2. This paper maintains high quality in its technical design, experimental setup, and result analysis, ensuring reliability and reproducibility.
3. The paper is exceptionally clear in its writing, organization, and visualization, making complex technical content accessible to a broad audience.

**Weaknesses:**

1. A core limitation of the work is its exclusive validation in simulated settings, which fails to account for the noise, uncertainty, and physical constraints of real-world scenarios, undermining its stated goal of "real-world applications in robotics and human-robot interaction."
2. This paper claims to support "multi-agent coordination," but its experiments and design fail to address scalability beyond 3 agents or dynamic obstacles with complex behaviors, limiting its utility for real-world settings like warehouses, hospitals, or crowded spaces.
3. This paper attributes LLM-Navi’s success to "zero-shot spatial reasoning inherent in LLMs" but provides no theoretical or empirical analysis of why LLMs excel at this task.

**Questions:**

Would removing obstacle tokens (only providing start/end points to the LLM) degrade performance? If so, by how much?

---

### Official Review · Reviewer_AUZo · 2025-11-01

**Soundness:** 2
**Presentation:** 3
**Contribution:** 2
**Rating:** 2
**Confidence:** 2

**Summary:**

The paper introduces LLM-Navi, a training-free pipeline that encodes floorplans, agent states, and trajectories as language tokens, enabling pretrained LLMs to perform zero-shot single- and multi-agent navigation. Core elements: (i) a unified token schema that represents trajectories as sparse anchor points connected by straight segments; (ii) a closed-loop refinement mechanism with two strategies—additive (replan from start) and compositional (continue from the last valid point); and (iii) an evaluation suite on real-world floorplan maps with SR/SPL/CR/WSR metrics, plus small demos for 2.5D/3D planning and humanoid motion. Results show strong zero-/few-turn planning across several LLMs and non-trivial multi-agent coordination on static, globally known maps. However, the current “dynamic” setting is effectively stepwise replanning on a static map, which does not reflect real dynamic environments where people move and layouts drift; the paper would benefit from time-aware baselines, VLM ablations, and robustness tests to map uncertainty.

**Strengths:**

* **Original framing:** A simple but effective anchor-token interface that makes spatial planning legible to LLMs and avoids rigid grid actions.
* **Closed-loop design:** Clear additive vs. compositional refinements with empirical comparisons; multi-turn improvements are demonstrated.
* **Breadth of zero-shot results:** Multiple LLMs tested; consistent gains without task-specific training.
* **Bridging potential:** Clean handoff from language plans to motion controllers, including 2.5D/3D and humanoid motion demos.
* **Clarity:** Pipeline and metrics are easy to follow; the evaluation set derived from real floorplans is reasonable and well described.

**Weaknesses:**

* **Dynamics are under-specified.**  The paper claims adaptation to “dynamic changes,” but these are not operationalized. Please distinguish moving obstacles (humans/carts), persistent layout edits (blocked doorway, moved furniture), and state-estimation errors (pose drift). Current experiments appear to use stepwise replanning on a static global map; it’s unclear which dynamics are actually handled.
* **No quantitative “real-time” evidence.** “Real time” is asserted but not measured. The paper should report a latency budget (sensing→LLM→controller), replan frequency (Hz), tokens/turn, and wall-clock per turn vs. map size and agent count, plus safety proxies (min time-to-collision (TTC), clearance).
* **Conflict resolution not benchmarked.** The pipeline is text-only. Given the spatial nature of the task, a multimodal LLM (VLM) that consumes the floorplan (or egocentric crops) as an image alongside the textual schema could reduce tokenization loss and improve narrow-passage/partial-obs performance. A small VLM ablation would clarify whether the tokenizer is a bottleneck.

**Questions:**

1. **Why text-only first?** What concrete advantages vs. VLMs (e.g., lower latency/cost, lower variance, easier safety controls, deterministic parsing)? Please quantify.
2. **Minimal VLM ablation.** Compare text-only to: (a) image-only floorplan; (b) late fusion (image + your text schema); (c) egocentric crops for partial-obs corridors. Report SR/SPL/CR, narrow-passage success, latency, and tokens.
3. **Scope of dynamics.** Which do you handle and evaluate: moving obstacles, persistent layout edits, pose drift? How are they instantiated?

---

### Official Review · Reviewer_iCWU · 2025-11-03

**Soundness:** 2
**Presentation:** 2
**Contribution:** 2
**Rating:** 2
**Confidence:** 3

**Summary:**

This paper introduces LLM-Navi, a LLM-based framework for path-finding in dynamic and cluttered environments. LLM-Navi enables robust spatial reasoning in realistic, multi-agent scenarios by uniformly encoding the environments( e.g.,real-world floorplans), dynamic agents, and their trajectories as tokens. The experiments show that LLMs can generate collision-free trajectories and resolve multi-agent conflicts in real time.

**Strengths:**

1. The idea of using LLM for spatial reasoning is interesting.
2. Well-written paper that is easy to read and follow.

**Weaknesses:**

1. The paper does not clearly demonstrate the necessity or advantage of employing LLMs for the classic pathfinding task. Since highly effective and deterministic algorithms (like A*) already exist for this problem, the unique value proposition of using an LLM remains unclear. To address this, the authors should include comparisons with classical planning algorithms to highlight the specific scenarios or capabilities where LLMs provide a distinct advantage, such as handling imperfect natural language instructions or reasoning under constraints that are difficult to formalize.
2. In constructing the test data, besides the optimal routes based on the A star algorithm, the authors appear to have employed more stylized prompts to evaluate the LLMs' reasoning capabilities in path planning. However, the paper lacks a systematic overview of these test scenarios. More importantly, it does not specify how the corresponding ground-truth labels were constructed for these prompts. A detailed description of this process is crucial for understanding the evaluation's validity and for ensuring the results are reproducible.
3. The current approach primarily relies on prompt engineering, which offers limited insights into the model's inherent reasoning capabilities for spatial planning. To deepen the work, it would be highly valuable to investigate the effect of fine-tuning LLMs on spatial planning datasets.

**Questions:**

1. The example of "hide and seek" in Figure 1 is interesting as it assigns a role to the path planner. It would be helpful if the authors can provide more examples like this and add further analysis, which will undoubtedly deepen the scope of the paper.
2. In which format the motion of humanoid generated, in joint space or motion of COM?

---

### Note · Authors · 2025-11-14

**Comment:**

I have read and agree with the venue's withdrawal policy on behalf of myself and my co-authors.

**Withdrawal Confirmation:**

I have read and agree with the venue's withdrawal policy on behalf of myself and my co-authors.